# Factors and Priorities Influencing Satisfaction with Care among Women Living with HIV in Canada: A Fuzzy Cognitive Mapping Study

**DOI:** 10.3390/jpm12071079

**Published:** 2022-06-30

**Authors:** Lashanda Skerritt, Angela Kaida, Édénia Savoie, Margarite Sánchez, Iván Sarmiento, Nadia O’Brien, Ann N. Burchell, Gillian Bartlett, Isabelle Boucoiran, Mary Kestler, Danielle Rouleau, Mona Loutfy, Alexandra de Pokomandy

**Affiliations:** 1Department of Family Medicine, McGill University, Montreal, QC H3S 1Z1, Canada; lashanda.skerritt@mail.mcgill.ca (L.S.); ivan.sarmiento@mail.mcgill.ca (I.S.); gillian.bartlett@health.missouri.edu (G.B.); 2Faculty of Health Sciences, Simon Fraser University, Burnaby, BC V5A 1S6, Canada; angela_kaida@sfu.ca (A.K.); margaritesanchez@gmail.com (M.S.); 3Chronic Viral Illness Service, McGill University Health Centre, Montreal, QC H4A 3J1, Canada; edenia1988@gmail.com; 4Viva Women, Vancouver, BC V5Z 0C9, Canada; 5Centre de Recherche du Centre Hospitalier de l’Université de Montréal, Montreal, QC H2X 0A9, Canada; nadia.obrien@phac-aspc.gc.ca (N.O.); danielle.rouleau@umontreal.ca (D.R.); 6Department of Family and Community Medicine, Faculty of Medicine, University of Toronto, Toronto, ON M5G 1V7, Canada; ann.burchell@unityhealth.to; 7Department of Family and Community Medicine, University of Missouri, Columbia, MO 65211, USA; 8Centre Hospitalier Universitaire Sainte-Justine, Université de Montréal, Montreal, QC H3T 1C5, Canada; isabelle.boucoiran@umontreal.ca; 9Oak Tree Clinic, Vancouver, BC V5Z 0C9, Canada; mary.kestler@cw.bc.ca; 10Women’s College Research Institute, Toronto, ON M5S 1B2, Canada; mona.loutfy@wchospital.ca; 11Research Institute of the McGill University Health Centre, Montreal, QC H4A 3J1, Canada

**Keywords:** HIV, women’s health, mixed methods, health equity, patient-centered care

## Abstract

Engagement along the HIV care cascade in Canada is lower among women compared to men. We used Fuzzy Cognitive Mapping (FCM), a participatory research method, to identify factors influencing satisfaction with HIV care, their causal pathways, and relative importance from the perspective of women living with HIV. Building from a map of factors derived from a mixed-studies review of the literature, 23 women living with HIV in Canada elaborated ten categories influencing their satisfaction with HIV care. The most central and influential category was “feeling safe and supported by clinics and healthcare providers”, followed by “accessible and coordinated services” and “healthcare provider expertise”. Participants identified factors that captured gendered social and health considerations not previously specified in the literature. These categories included “healthcare that considers women’s unique care needs and social contexts”, “gynecologic and pregnancy care”, and “family and partners included in care.” The findings contribute to our understanding of how gender shapes care needs and priorities among women living with HIV.

## 1. Introduction

In Canada, access to care and antiretroviral therapy (ART) has transformed HIV into a manageable chronic condition. Despite these advances, vast inequities in healthcare access, retention in care, HIV clinical outcomes and overall health and survival for people living with HIV persist [1,2]. In particular, although women represent one-quarter of people living with HIV in Canada [3], HIV programming has not adequately responded to their unique and evolving social and health priorities across the life course [4]. The lack of services that respond to the needs and priorities of women living with HIV has manifested in worse HIV-related outcomes among women compared to men across the HIV care cascade, including lower engagement in care, worse ART adherence and shorter life expectancy [1,2]. Gendered health inequities are further exacerbated by healthcare services that overlook the specific care needs and priorities of women living with HIV, resulting in delayed cervical and breast cancer screenings, missed opportunities for family planning counselling [5,6,7], and neglect of other sexual and reproductive health priorities [7]. These gaps in care are corroborated by women’s description of feeling overlooked in healthcare because of a lack of women-specific services [4].

Underlying the gendered inequities in HIV care and outcomes are multiple complex social and structural factors. To best meet women’s health care needs, healthcare services transformation should be anchored in the gendered priorities that shape patient satisfaction with care. A literature review synthesizing findings from articles reporting perceptions, experiences of, or satisfaction with healthcare services among people living with HIV identified seven themes [8]: (1) healthcare provider-patient relationship, (2) HIV specialist knowledge, (3) continuity of care, (4) health service access, (5) access to information and support, (6) co-ordination between healthcare providers, and (7) patient involvement in decision-making. The review did not, however, examine the role of gendered health considerations in satisfaction with care or the relative importance of different features of care by gender. This gendered lens is critical as perceptions of the clinic environment and care providers impact engagement with care among women living with HIV [9]. Meanwhile negative experiences, including experiencing or anticipating stigma and discrimination, are associated with poorer engagement in HIV care and adherence to ART [10,11]. Women’s perceptions of their care environments affect not only HIV outcomes, but also sexual and reproductive health, an aspect of care particularly neglected. Women living with HIV who report feeling comfortable with healthcare providers are more likely to discuss their reproductive goals with their care providers [12]. Although frameworks recognizing the need for gendered care approaches exist [13,14], few studies have examined the care preferences and priorities of women living with HIV. Integrating these priorities and evidence-based care recommendations can maximize the care engagement, health and wellbeing of this population [15]. Understanding patient priorities and the causal relationships between aspects of care and satisfaction can also inform the implementation of interventions.

Satisfaction with care is a useful construct as it captures expectations of and experiences with healthcare [15]. Features of care that influence satisfaction from the perspective of women living with HIV remain poorly understood. Re-examining these features of satisfaction with care from the perspective of women living with HIV and determining the relative importance of these considerations can inform healthcare delivery that better aligns with their needs and redresses the historic marginalization of women in the design and delivery of HIV care [16]. The aim of this study was to identify relevant factors influencing women’s satisfaction with HIV care, the relative importance of these factors to women living with HIV, and the causal relationships between these factors and satisfaction. 

## 2. Materials and Methods

### 2.1. Theoretical Framework

This study was guided by the social determinants of women’s health framework described by McGibbon and McPherson [17]. Through this framework, the social determinants of women’s health inequities are understood by bridging theories of social determinants of health, intersectionality theory and complexity theory. These theoretical frameworks informed our use of a mapping study design to capture and analyze the compounding effects of gender, oppression and power systems on healthcare priorities and experiences with healthcare. Given its known associations with care engagement, health outcomes, and patient experience, we considered patients’ satisfaction with care as a critical indicator of appropriate health service delivery. We aimed to center the priorities of women, acknowledging the role of gender and power dynamics in shaping the healthcare experience. The framework informed our interpretation of the identified factors influencing satisfaction with care. When examining each factor, we considered the role of gender and its intersections on women’s healthcare expectations and perceptions. 

### 2.2. Participatory Research Approach

We applied a participatory research approach, which aims to meaningfully involve community members in research impacting their community by responding to their priorities and drawing on their strengths and knowledge [18]. In this study, we considered women living with HIV as experts in their experiences with their HIV care [16]. Our research approach grounded existing evidence in the experiential expertise of women living with HIV. Two Peer Research Associates (PRAs), women living with HIV with research training and experience were partners throughout the research process, including the study design, interview guide development, mapping facilitation and data collection, data analysis, and interpretation of findings. 

### 2.3. Fuzzy Cognitive Mapping

We used Fuzzy Cognitive Mapping (FCM) [19] to identify factors influencing satisfaction with HIV care among women living with HIV in Quebec and British Columbia, Canada. FCM is a systematic process of knowledge creation that generates a concept map representing stakeholder understanding of causal relationships. The resulting map is a directed graph with nodes and numerically weighted arrows that model causes, effects, and the strength of causal relationships. FCM is a commonly used tool for stakeholder engagement in environmental studies and urban planning [20,21] and has been used in participatory health research to capture Indigenous knowledge [19,22,23], and to ground published knowledge and literature reviews in stakeholder understanding [24]. The use of visual language in FCM supports its use across different languages, cultures, and education levels. In this study, we used FCM to collate different sources of causal understanding, namely literature evidence and stakeholder knowledge.

### 2.4. Study Participants and Recruitment

A combination of convenience and purposive sampling was used to recruit participants to this study. Women living with HIV who were 18 years of age and older, spoke English or French, and were living in the study provinces of Quebec or British Columbia were eligible to participate. PRAs [25] and study investigators approached potential participants from their networks, through clinics, and contacted previous participants of the Canadian HIV Women’s Sexual and Reproductive Health Cohort Study (CHIWOS), a community-based participatory research project that followed women living with HIV in Canada between 2015 and 2018 [26]. Participants were contacted by phone, email, or in person and invited to participate in the mapping sessions. We recruited participants of diverse ages, immigration experiences, rural and urban geographic locations, and reproductive health experiences (previous pregnancies, pregnancy intentions, menopausal status, and current contraceptive use). Recruitment took place between December 2020 and March 2021. We stopped recruitment when data saturation was reached, defined by the research team as the point at which no additional factors arose.

This study received ethics approval from the McGill University Health Centre, the Simon Fraser University, and the University of British Columbia Research Ethics Boards. All participants provided informed verbal consent. We provided a 25 CAD honorarium to interview participants and an additional 50 CAD honorarium to participants who attended the follow-up group discussion. An overview of the study process is illustrated in Figure 1.

### 2.5. Data Collection

Step 1. Constructing the literature-based map

We presented a synthesis of published evidence in the form of a map that women living with HIV used as an initial common ground to discuss factors they considered relevant to their satisfaction. The lead author constructed a map of themes influencing satisfaction with HIV care among people living with HIV based on findings from a mixed-studies review by Cooper et al., published in 2016 [8]. This review synthesized evidence from 23 studies (*n* = 2719) exploring aspects of care associated with satisfaction among people living with HIV in Australia, Canada, the Netherlands, the UK, and the USA between 1996 and 2015. Men living with HIV represented 61% of participants of the included studies reporting gender. The review, however, did not examine gendered care priorities or how gender shapes satisfaction with HIV care. For this project, a map summarizing themes from the literature review served as a starting point for women living with HIV to build their individual map following a standard protocol described below. The literature-based map presented factors and unweighted directed pathways toward satisfaction with HIV care.

Step 2. Fuzzy Cognitive Mapping sessions

Through individual interviews, participants adjusted the literature-based map to represent their personal perspective. A PRA and the lead author co-facilitated the individual mapping sessions in French or English per participant preference. Each participant completed a brief demographic questionnaire. The PRA guided the interview and mapping process, while the lead author illustrated the map in real time and took notes of the discussion. The facilitators explained the process of Fuzzy Cognitive Mapping to the participant, presented the literature-based map, guided participants in developing their map, using a predefined qualitative interview guide. The guide invited women to first reflect on the healthcare they had received or were receiving and define satisfaction with HIV care. “*In this interview, we would like you to reflect on that healthcare that you’ve received or are receiving. Was it OK? Was it the right care for you? Is it the right care for other women like yourself? If not, what could have made it better?”*

The PRA facilitator then guided participants in the stepwise construction of their map. First, women were asked an open-ended question to elicit their understanding of the outcome, satisfaction. “*What does satisfaction with HIV care mean to you?*” The PRA facilitator asked women to adjust the map by adding or removing factors that positively or negatively influence their satisfaction with HIV care. “*What factors do you believe positively or negatively influence your satisfaction with HIV care?*” After arriving to a final list of factors, the PRA facilitator asked women to identify all the causal relationships between factors. “*For each factor, one at a time, please identify all the causal relationships between that factor and others (meaning this factor causes another factor or vice-versa).*” The PRA facilitator asked women to assign a weight between −5 and +5 indicating the strength and direction of the relationship. “*Please assign a weight between −5 and +5 indicating how strong the relationship is and whether the influence is positive or negative*.” Positive weights indicated that an increase in the originating factor would increase the landing factor, and negative weights the opposite. Values closer to −5 and +5 indicated stronger influences, and values closer to 0 corresponded to weaker ones. At the end of the interview, the PRA facilitator asked women to review their Fuzzy Cognitive Map and make any necessary changes before confirming the final version. “*Is there anything you would like to change in your final map?”*

We used the graph editing software yED [27] to draw the maps in real time during the interviews. Interviews were conducted virtually over the videoconferencing platform Zoom. Interview content was audio-recorded and transcribed verbatim. The lead author also took notes during each interview to capture important observations and meanings around participants definition of factors and their relationships. After each interview, a reflexive debrief was conducted between the mapping co-facilitators to note the interview experience and any issues relevant to the data collection process. Individual interviews varied in length from 35 min to 2 h.

### 2.6. Data Analysis and Interpretation

#### 2.6.1. Qualitative Analysis of the Listed Factors Influencing Satisfaction with HIV Care

FCM yielded both qualitative and quantitative data. Following methods described by Erlingsson and Brysiewicz [28], we conducted qualitative content analysis to condense the factors from the individual maps into categories. The lead author developed the first level of aggregation using a pattern correspondence table to collapse factors of similar meaning across individual maps [19]. For example, the factors “family and partners included in care”, “mom is included in care”, “family involved in care at the clinic”, “care and resources at clinic extended to family”, were standardized as “family and partners included in care”. This process of standardizing factors was ongoing and iterative. The final three participants identified factors consistent with those in the previous 20 interviews. We considered data saturation reached and stopped the data collection after 23 interviews. The lead author and two PRAs then conducted content analysis identifying categories from the standardized factors. The categorization of factors and their meanings were validated with input from other members of our multidisciplinary research team members, which included HIV clinicians and researchers. 

#### 2.6.2. Quantitative Analysis of Assigned Weights Representing the Strength of Associations

The individual maps were first presented as individual adjacency matrices. The matrices had an equal number of columns and rows, with each row and column corresponding to the standardized factors identified across all the maps. Each matrix cell corresponded to an arrow connecting two factors, from the row to the column. The cell value equaled the weight of that arrow assigned by the participant in the corresponding map. Cells had a zero value if the relationship was not on the map. We normalized the weights by dividing them by 5 to put them in the −1 to +1 range. We analyzed each participant’s matrix using Transitive Closure, a tool derived from graph theory and available in Python version 3.7 and CIETmap [29]. Transitive closure recalculates the matrices by identifying the effects that one factor has on others through direct and indirect paths. The resulting transformed matrix specifies the strongest influence between nodes across the direct and indirect paths, where the strength of an indirect path equals the strength of the weakest relationship in the pathway. 

From the 23 transformed matrices, we calculated the average weight of factor-level relationships. We used the categorization from the content analysis to condense the factor-level relationships into a category-level matrix. The category-level weights equaled the relative sum of the average weights of factor-level relationships with that category. The weight of the relationship between categories equaled the sum of the mean weights from each factor in the outgoing category to each factor in the incoming category. The sum was then divided by the maximum cumulative weight across all categories in the average matrix to obtain relative values between −1 and +1. 

#### 2.6.3. Network Analysis

In addition to calculating the strength of the influences on satisfaction with HIV care, we also described the categories using indegree and outdegree centrality. For each category, we summed the absolute weights of incoming (indegree) and outgoing (outdegree) arrows as measures of centrality. A higher sum would indicate a higher centrality measure.

#### 2.6.4. Member-Checking Group Discussions to Validate Results

We conducted three virtual member checking [30] group discussions with nine mapping participants to establish trustworthiness in the results. They confirmed their agreement with the analysis results and provided additional interpretations of the categories and their relative importance. These discussions contributed to the final framing of the research findings and recommendations. 

## 3. Results

Of 26 women living with HIV approached to participate, 23 agreed to participate in the individual mapping sessions. Table 1 presents the characteristics of study participants. Median age was 47 years [IQR: 19, 66], with a median 20 years living with HIV. The majority of women identified as cisgender, heterosexual/straight, white, with completed post-secondary education and did not live with a partner. Two-thirds of women had a household income above $20,000 CAD. Nine (39%) participants had a pregnancy since their HIV diagnosis, 17% intended to become pregnant in the future, and 26% reported any contraception use in the last six months (including hormonal contraceptives, long-acting reversible contraceptives, male/female condoms, or other methods). 

### 3.1. Factors Influencing Satisfaction with HIV Care

The literature-based map included seven factors associated with satisfaction with HIV care (Figure 2). Women adjusted this map, identifying 79 factors influencing satisfaction with HIV care directly or indirectly and 1083 relationships (arrows) connecting these factors. The 79 factors included those from the literature-based map and new factors added by participants. We condensed the 79 factors into 10 categories. In Table 2, we present the final classification of factors in each category, including factors retained from the literature-based map and factors added by participants. “Accessible and coordinated services” and “healthcare provider expertise” were themes from the literature-based map retained in the final map but were further expanded by additional factors added by participants. The remaining eight categories reflected constructs with new or different meanings than those from the literature review. The final categories were (1) feeling safe and supported by healthcare providers (HCPs) and clinics, (2) accessible and coordinated services, (3) healthcare provider expertise, (4) care that considers women’s unique care needs and social contexts, (5) empowerment/self-care/self-advocacy, (6) focus on mental well-being, (7) peer support and community involvement in care, (8) gynecologic and pregnancy care, (9) inclusion of family and partners in care, (10) care that adapts with aging.


Figure 3 presents the summary map with the ten final categories influencing satisfaction with HIV care, the highest weighted category-level relationships, and weights after transitive closure. The final map shows the two categories that closely reflected themes from the literature-based map as well as the categories that were distinct from initial themes presented in the literature. “Feeling safe and supported”, “Accessible and coordinated services”, and “Healthcare provider expertise” most strongly influenced satisfaction directly, while the remaining categories primarily influenced satisfaction indirectly, through their effects on “Feeling safe and supported.”

Table 3 lists the category weights on satisfaction with HIV care and their relative weighted importance in the scale −1 to +1. The weights account for both direct and indirect pathways through which the categories influence satisfaction with HIV care.

#### 3.1.1. Feeling Safe and Supported by Healthcare Providers and Clinics

Providers’ and clinics’ attention to patients’ feelings of safety and support had the most important influence on satisfaction with HIV care according to participants. The category “Feeling safe and supported by HCPs and clinics” included 26 factors (Table 2) corresponding to the care environment and approaches centering patients’ need to feel secure and cared for with kindness, compassion, and without judgement. The category combined constructs retained from the literature-based map, namely, good relationships with HCPs, receiving information and support, confidentiality, and sensitivity to stigma. Participants also added new factors, such as the importance of genuine support, culturally sensitive care, and the absence of judgement for cancelling clinic appointments. Many factors added by the participants reflected the gender-specific experience or needs, such as participants’ preference for female HCPs, not being judged for pregnancies or lifestyle, and having a welcoming, family-friendly, trans-inclusive waiting room in the clinic.

“Feeling safe and supported” directly influenced satisfaction with HIV care but, as Figure 3 shows, it also functioned indirectly by increasing access to services. The strong internal dynamics between factors in the category are reflected by a self-pointing arrow from “feeling safe and supported” onto itself. Figure 4a illustrates the factor-level relationships within “feeling safe and supported”. These internal dynamics were mainly attributed to the reinforcing relationships between five factors: (1) Having good/excellent relationships with HCPs; (2) receiving information and support from HCPs and clinics; (3) confidentiality and sensitivity to stigma; (4) follow-up from HCPs and clinic; and (5) feeling heard during appointments. The two most influential factor-level relationships were the effect of receiving information and support and confidentiality and sensitivity to stigma on having excellent relationships with healthcare providers. 

#### 3.1.2. Accessible and Coordinated Services

“Accessible and coordinated services” had the second strongest influence on satisfaction. The category consisted of 17 factors (Table 2). In identifying these factors, participants discussed the importance of timely and coordinated care and treatment, particularly when accessing care from multiple services and providers. Participants identified communication between providers and access to various services at one site as important for avoiding stigmatizing experiences in non-HIV specialized healthcare settings.

Strong internal dynamics were observed within this category, as depicted in Figure 4b by the self-pointing arrow. These internal dynamics were mainly attributed to the important relationships between four factors: (1) accessibility of healthcare, (2) coordinated healthcare services, (3) having healthcare providers who communicate with each other, and (4) having resources and care located in one place. The most influential factor-level relationship within this category was the effect of coordinated healthcare services on the accessibility of healthcare. Participants discussed the importance of accessibility and coordination to reduce or mitigate barriers to services and treatment.

#### 3.1.3. Healthcare Provider Expertise

The category “healthcare provider expertise” was composed of three factors, (1) provider expertise in HIV, (2) having a pharmacist as part of the care team, and (3) adequate training of nurses and doctors (including knowledge of women’s health and reproductive health in the context of HIV). The third factor capturing participants challenges in receiving both HIV and equitable women’s health care. As shown in Figure 3, “healthcare provider expertise” influenced satisfaction directly and indirectly through its effect on women’s feelings and safety and support. Women cited their challenges accessing quality care in settings outside of HIV-specialized clinics, where healthcare providers had less knowledge about women’s particular care needs and stigma concerns.

#### 3.1.4. Additional Categories Influencing Satisfaction with HIV Care

In addition to “Feeling safe and supported” the final map contained seven other categories that were distinct from constructs in the literature-based map. Women cited “*empowerment and self-care”* as influencing satisfaction by contributing to their self-advocacy and consequently having their specific needs and concerns met. Women identified care that “*considers women’s unique health needs and social contexts”* as important to their satisfaction, describing care approaches and treatments supported by research on HIV and women, and that also considers patient health and social needs and concerns that are specific to women. Care that “*focuses on mental well-being”* included mental health and social services available at clinics or access to mental health experts with knowledge on HIV to address trauma and violence. “*Peer support and community involvement in care”* referred to peer vetted referrals to services, peer and social support groups, and the meaningful involvement of women living with HIV and community in health service decision-making. “*Gynecologic and pregnancy care”* captured the importance of receiving gynecologic care, supportive pregnancy care and support for pregnancy decisions, including termination. In describing the importance of “*family and partners included in care”* women described clinics where their young children could also receive care and where partners or family members can access resources and information. The final important category for participants was *“care that adapts with aging”*. This category includes aging and changes in cognitive function that accompany aging.

### 3.2. Network Analysis

“Feeling safe and supported” had the highest indegree centrality, meaning it was the most important effect among all the relationships in the map both for the number of arrows pointing toward it and the strength of those arrows (indegree centrality score of 2.78). The second most important effect in the category map was “accessible and coordinated services” (indegree centrality score of 1.47). The importance of these two categories as relevant effects on the maps was even higher than the importance of the main outcome “satisfaction with HIV care” (indegree centrality score of 1). Outdegree centrality also identified “feeling safe and supported” and “accessible and coordinated services” as the most important influences, not only on satisfaction but also on other categories on the map.

## 4. Discussion

In this study, we used a participatory research approach to identify factors influencing satisfaction with HIV care and examine the relative importance of these factors from the perspective of 23 diverse women living with HIV in two Canadian provinces. Ten categories influencing satisfaction with HIV care were identified. “Feeling safe and supported”, “accessible and coordinated services”, and “healthcare providers’ expertise” most strongly influenced satisfaction and captured constructs consistent with the literature. In addition, women living with HIV identified, defined, and elaborated on social and health considerations that also shape their satisfaction with care. These categories were care that considers women’s specific needs and social contexts, empowerment/self-care/self-advocacy, care that focuses on mental well-being, peer support and community involvement in care, gynecologic and pregnancy care, the inclusion of family and partners in care, and finally care that adapts with aging. Six out of ten categories included gender-specific factors, despite the interview guide not specifically asking how women perceived their gender to affect their care. This finding highlights the importance of gendered considerations in shaping women’s care needs and satisfaction with healthcare delivery. Understanding the features of care that influence women’s satisfaction, their relative importance and causal mechanisms is essential for designing services that meet women’s care needs. Our study was unique in its novel use of FCM to integrate existing literature and the experiential expertise of women living with HIV. Our results contribute to the literature on HIV care delivery by presenting a complex picture of how women-specific factors shape the healthcare priorities of a sample of women living with HIV.

Feeling safe and supported in healthcare settings was the most important and central consideration in participants’ satisfaction with HIV care. Although this category included constructs consistent with the existing literature [8], participants elaborated further, identifying additional factors that define feelings of safety and support in healthcare. Participants weighted communication of information and resources and confidentiality and sensitivity to stigma as contributing strongly to their relationships with their healthcare providers. These findings align with previous studies that identified patient-provider communication as important for care engagement [31,32] and resilience, particularly for more marginalized populations living with HIV [33]. Interpersonal aspects of care that respond to known structural barriers to care for women living with HIV also emerged as significant. Participants described the importance of provider and clinic sensitivity to stigma, a known barrier to accessing HIV care [34]. Fear of HIV-related stigma has been found to impact interactions with HIV clinics and staff [35]. The importance of patient-provider relationships, including those developed with female healthcare providers, has been shown to affect health outcomes, such as discussing reproductive goals with healthcare providers and sexual and perinatal wellbeing among women living with HIV [12,36]. Healthcare provider gender has been shown to predict health outcomes outside of HIV care [37,38]. Women living with HIV who have good relationships with their care providers may be more likely to remain engaged in and feel safe, supported, and satisfied with their HIV care. 

Accessible and coordinated services and healthcare provider expertise represented the second and third most important considerations in women’s satisfaction with HIV care. These considerations are consistent with the literature on aspects of care valued by people living with HIV [8]. Clinic distance, long wait times, and inconvenient clinic hours have been cited as barriers to healthcare access in studies with people living with HIV [39]. Engagement in HIV care is lower among women compared to men [40]. Through FCM, women in this study described how the coordination of services, communication between providers and the provision of multiple services at one clinic increase their access to care. Consistent with previous research, participants also identified healthcare provider expertise as important to their satisfaction. Healthcare for people living with HIV includes HIV management but also the management of multiple health conditions, co-morbidities, and reproductive health [41]. The complexity of care considerations requires multidisciplinary approaches, often including non-HIV specialized care providers and clinics. Stigma in non-HIV specialized clinics is a known barrier to accessing services [4,31]. Stigma contributes to the avoidance of healthcare, delays accessing treatment, and treatment nonadherence among people living with HIV [42,43,44,45,46]. Women living with HIV identified the importance of care providers with adequate knowledge of HIV to prevent enacted stigma in healthcare encounters. 

Women living with HIV described gendered health considerations that shape their satisfaction. Gendered considerations emerged in larger categories influencing satisfaction such as women feeling safe and supported. Women identified female healthcare providers and the absence of judgment of sexual and reproductive health behaviors as contributing to their feelings of safety and support. While “feeling safe and supported” included gender-specific factors, other categories were completely shaped by gender. Women’s unique needs and social contexts emerged as a category defined by gender. Previous literature has described how gender can inform women-specific HIV services and aspects of HIV care [13]. In this study, women living with HIV described the need for healthcare to recognize their gendered social contexts as with unique and complex histories. Women outlined the multiple mechanisms through which their social contexts influence satisfaction, including through empowerment, mental health, peer and community support needs, the involvement of family or partners in care, gynecologic and pregnancy care needs, and the evolution of care across the life course. These considerations should be interpreted through an intersectional lens, recognizing that health priorities are shaped by diverse and overlapping social and structural factors [47,48,49]. Gender, socioeconomic, and racial inequities intersect to inform patient healthcare needs and interactions with healthcare providers. These inequities also give rise to gaps in research on the care priorities of women living with HIV [50]. Given the prevalence of intersectional stigma in women’s experiences navigating HIV healthcare, [10,11,51] healthcare must go beyond service provision. According to women living with HIV in our study, satisfying care must also responds to the gendered social contexts of women living with HIV.

### 4.1. Limitations

We did not collect data on clinical health outcomes or engagement in HIV care. Our sample did not include trans women; therefore, the findings may not capture the unique care priorities among this population. Future research should examine how gender shapes satisfaction with care among people living with HIV of diverse genders. Our literature review synthesis and construction of the literature-review map did not include PRA perspectives. Through mapping interviews, we aimed to integrate the perspectives of women living with HIV in existing literature. Data collection was limited to women living with HIV in two Canadian provinces, but the social contexts and demographics, including immigration status and racial identity of women living with HIV differ significantly in these two provinces [26]. Additionally, the study took place within the context of universal healthcare delivery. Findings may not extend to other settings, particularly where healthcare is delivered through different models, such as private delivery. 

### 4.2. Strengths

Our recruitment strategy aimed to engage structurally marginalized women and underserved communities. Around half of participants identified as non-white, around one-fifth identified as a gender minority, and 30% as low income, reflecting diversity in socioeconomic status. In this study, we asked women living with HIV to update existing literature to reflect factors influencing their satisfaction and to weigh the importance of relationships between factors. Although previous studies identified aspects of care valued by people living with HIV, our findings illustrated how gendered health and social considerations shape women’s satisfaction with HIV care. Our results and interpretations are grounded in women’s understanding of the causal relationships between their care priorities and satisfying HIV care.

## 5. Conclusions

Women living with HIV emphasized the importance of care approaches that increase their feelings of safety and support. The role of gender in shaping satisfaction with HIV care supports the view that all healthcare should be provided through a person-centered approach that considers gender along with other health and social contexts, which additionally shape the healthcare experience. The findings of this study revealed the importance of relationship- and trust-building with healthcare providers as a priority for women living with HIV, consistent with principles of person-centered and women-centered HIV care [4,52]. Efforts to improve HIV care and health outcomes for women living with HIV must integrate women’s gendered priorities and perspectives in designing and delivering health services [53]. Our findings illustrate a systematic approach to developing conceptual maps grounded in the literature and stakeholder expertise. The results can guide future research and health services that consider the role of gender and its intersection in the care needs and priorities of women living with HIV. 

## Figures and Tables

**Figure 1 jpm-12-01079-f001:**

Study process overview.

**Figure 2 jpm-12-01079-f002:**
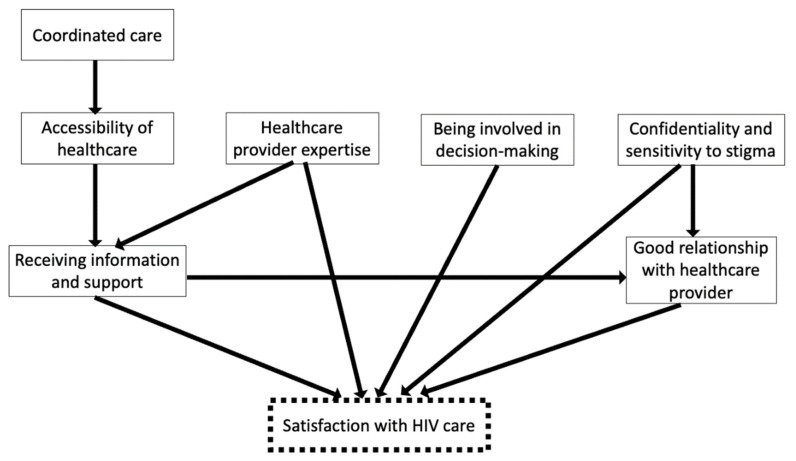
Literature-based map synthesizing findings from Cooper et al. on the factors that influence satisfaction with HIV care among people living with HIV.

**Figure 3 jpm-12-01079-f003:**
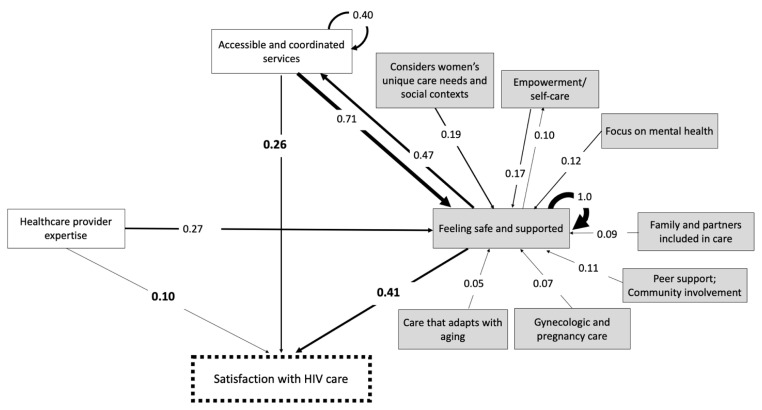
Summary Fuzzy Cognitive Map of category-level relationships showing the three highest weighted direct influences and the highest weighted indirect influences on satisfaction with HIV care. Weights closer 1 indicate stronger influences. The highest weighted influences on satisfaction with HIV care are bolded. Grey boxes represent new constructs added to the literature-based map.

**Figure 4 jpm-12-01079-f004:**
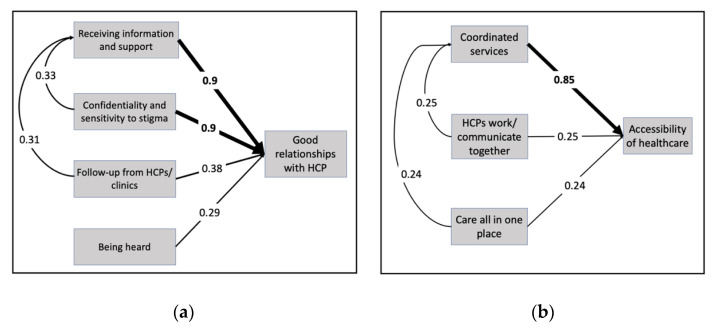
Self-pointing arrows in the final map at the factor level. The strongest internal dynamics between factors within the categories (**a**) feeling safe and supported and (**b**) accessible and coordinated services. Weights closer to 1 indicate higher influences. The highest weighted influences are bolded.

**Table 1 jpm-12-01079-t001:** Participant characteristics.

Characteristics	Overall (*n* = 23)
Age, Median [Min, Max]	47 [Min: 19, Max: 66]
Years living with HIV, Median [Min, Max]	20 [Min: 10, Max: 36]
Gender	
Cis woman	21 (91.3%)
Genderqueer	2 (8.7%)
Ethnicity	
African/Caribbean/Black	8 (34.8%)
Indigenous	2 (8.7%)
Asian	1 (4.3%)
White	12 (52.2%)
Sexual orientation	
Bisexual/Lesbian/Queer	5 (21.7%)
Heterosexual	18 (78.3%)
Relationship status	
Married/Common-law/In a relationship	9 (39.1%)
Single/Separated/Divorced/Widowed	14 (60.9%)
Education	
Post-secondary or higher	16 (69.6%)
Secondary or lower	7 (30.4%)
Household annual income, <20,000 CAD	7 (30.4%)
Pregnancy since HIV diagnosis	9 (39.1%)
Intends to become pregnant in the future	4 (17.4%)
Contraception use in last 6 months	6 (26.1%)
Post-menopause	11 (47.8%)

**Table 2 jpm-12-01079-t002:** Categorization of factors influencing satisfaction with HIV care among women living with HIV. Factors retained from the literature-based map in grey.

Final Category	Factors
Feeling safe and supported by HCPs and clinics	**1. Good/excellent relationships with HCPs**
	**2. Receiving information and support from HCPs and clinics**
	**3. Confidentiality and sensitivity to stigma**
	4. Access to genuine support
	5. Follow-up from HCPs and receptionists
	6. Culturally sensitive care
	7. Caring, kind, genuine social worker/support, nurse practitioner
	8. Reminders for appointments outside of clinic
	9. Dentist comfortable treating people with HIV
	10. Outreach workers at clinic
	11. Building relationships with healthcare team
	12. Honesty from HCP; trusting relationship
	13. HCP advocacy
	14. Female HCP
	15. Respect from HCP and community
	16. Not being treated differently when accessing non-HIV care
	17. Using kind and considerate language
	18. Continuity with HCP and social workers
	19. Not being judged for pregnancies or lifestyle
	20. HCP being good listeners; feeling heard; Questions and concerns being addressed
	21. Regular monitoring of CD4 count and viral load
	22. No judgement from reception when cancelling appointments
	23. Welcoming, family-friendly, trans-inclusive waiting room
	24. Addressing side effects of ARTs
	25. Disclosure of HIV status to HCPs
	26. Less confidentiality in rural settings
**Accessible and coordinated services**	**1. Accessibility of healthcare**
	**2. Coordinated healthcare services**
	3. Receiving HIV medication while incarcerated
	4. HCPs that work as a team; communicate with each other
	5. No waitlist to access clinic services
	6. COVID-19 pandemic interfering with access to services
	7. Easy transportation to/from appointments
	8. Resources and care all in one place (Holistic)
	9. Accessing services over the phone/remote
	10. HCP going above and beyond to be accessible
	11. HCP available for non-HIV healthcare resources
	12. Close geographical distance/transportation to/from clinic
	13. Being able to see a doctor
	14. Clinic ensuring primary care is happening
	15. Links to non-HIV specialists
	16. Complementary healthcare rather than medication
	17. No delayed access to ARTs due to lack of insurance for immigrants
**Healthcare provider expertise**	**1. HCP with expertise in HIV**
	2. Pharmacists essential part of care team; prevent drug interactions, manage side effects
	3. Adequate training of nurses and doctors (incl. HIV, women’s health, and reproduction)
Empowerment/self-care/self-advocacy	**1. Collaborative approach between me and my HCP**
	2. COVID-19 pandemic interfering with spiritual practices
	3. Feeling empowered by HCP to self-advocate
	4. Having the option in advance to refuse or accept trainee HCPs in appointments
	5. Training on self-care
	6. Exercise as part of care
	7. Celebration of health milestones
	8. No doom and gloom attitude
	9. Belief in self and prayer
	10. Patience and confidence in myself and my strength
Care that considers women’s unique care needs and social contexts	1. Considers the social contexts of women living with HIV
	2. Person-centered care
	3. HCP focused on my needs and concerns
	4. Research on HIV and women
	5. Access to women-specific treatments
	6. HCP considers my history and context
	7. Care that adapts to my unique needs
	8. Care that considers my health in the context of the COVID-19 pandemic
Focus on mental well-being	1. Mental health and social services integrated in clinic
	2. HIV-knowledgeable psychiatrist
Peer Support; Community involvement in care	1. Peer vetted referrals to non-discriminatory services
	2. COVID-19 pandemic interfering with social support
	3. Community/peer support groups
	4. Collaborative approach between medical and community
	5. Meaningful Involvement of Women Living with HIV/AIDS (MIWA)
	6. Disability insurance as barrier to community engagement & support
Gynecologic and pregnancy care	1. Receiving gynecologic care
	2. Supportive pregnancy care
	3. Being given options during pregnancy (e.g., abortion)
Inclusion of family and partners in care	1. Inclusion of family and partners in care
Care that adapts with aging	1. Healthcare that adapts with aging
	2. Focus on cognitive function changing with aging

In bold: Factors retained from the literature based map.

**Table 3 jpm-12-01079-t003:** Categories influencing women’s satisfaction with HIV care and their relative category weightings considering direct and indirect pathways.

Category	Weight
1. Feeling safe and supported by HCPs and clinics	0.41
2. Accessible and coordinated services	0.26
3. Healthcare provider expertise	0.10
4. Care that considers women’s unique care needs and social contexts	0.06
5. Empowerment/self-care/self-advocacy	0.05
6. Focus on mental well-being	0.03
7. Peer Support; Community involvement in care	0.03
8. Gynecologic and pregnancy care	0.03
9. Inclusion of family and partners in care	0.02
10. Care that adapts with aging	0.02

## Data Availability

Data are available upon request to corresponding author.

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
