# Peer review of "Factors and Priorities Influencing Satisfaction with Care among Women Living with HIV in Canada: A Fuzzy Cognitive Mapping Study"

_jpm, 2022, doi:10.3390/jpm12071079_

Round 1
Reviewer 1 Report
The paper is extremely innovative in its approach of identifying factors influencing satisfaction with HIV care. The approach of Fuzzy Cognitive Mapping in particular is novel. There are a few minor revisions that I would recommend:
1. Error! Reference source not found was on page 6 line 253. I assume this is a typo.
2. It appears that the authors have collapsed categories on page 5 line 207-209; the process is collapsing.
3. It appears more apt that lines 265-271 on page 7 are moved up to the methods. This brings more clarity about the relationship between literature and the FCM interviews in addition to the collapsing process.
4. The theoretical framework needs to be strengthened - did it help shape the design of the study or the aims of the study? Seems like the latter. How did the theoretical framework shape the results of the study.
5. It would be helpful to know whether the synthesis of the literature review were done with the assistance of the 2 RAs who were community partners. If not, that may be a limitation where the RAs were involved later on.
Author Response
The paper is extremely innovative in its approach of identifying factors influencing satisfaction with HIV care. The approach of Fuzzy Cognitive Mapping in particular is novel. There are a few minor revisions that I would recommend:
Response: Thank you the constructive feedback on our manuscript. We have made revisions according to the comments below and indicated these revisions in tracked changes.
- Error! Reference source not found was on page 6 line 253. I assume this is a typo.
Response: Thank you for pointing out this typo. We have corrected the error.
- It appears that the authors have collapsed categories on page 5 line 207-209; the process is collapsing.
Response: Thank you for identifying this process as collapsing. We have now included the correct terminology. The text now reads
The lead author developed the first level of aggregation using a pattern correspondence table to collapse factors of similar meaning across individual maps [19].
- It appears more apt that lines 265-271 on page 7 are moved up to the methods. This brings more clarity about the relationship between literature and the FCM interviews in addition to the collapsing process.
Response: Thank you for this suggestion. We have moved the lines describing the length of the participant interviews to the methods section. We have removed the lines describing the use of the literature-based map as a starting point for individual mapping from the results section, as this was already described in the methods sections. We feel that this now clarifies the relationship between literature and the FCM interviews while eliminating redundancy between these two sections.
- The theoretical framework needs to be strengthened - did it help shape the design of the study or the aims of the study? Seems like the latter. How did the theoretical framework shape the results of the study.
Response: Thank you for this comment. We have revised the theoretical framework description to highlight how the theoretical framework informed the study design, aims and interpretation. We have emphasized that our framework informed our decisions to use a mapping approach for data collection. We have described how our use of this framework informed our interpretation of study findings. This section of the manuscript now reads:
This study was guided by the social determinants of women’s health framework described by McGibbon and McPherson [17]. Through this framework, the social determinants of women’s health inequities are understood by bridging theories of social determinants of health, intersectionality theory and complexity theory. These theoretical frameworks informed our use of a mapping study design to capture and analyze the compounding effects of gender, oppression and power systems on healthcare priorities and experiences with healthcare. Given its known associations with care engagement, health outcomes, and patient experience, we considered patients’ satisfaction with care as a critical indicator of appropriate health service delivery. We aimed to centre the priorities of women, acknowledging the role of gender and power dynamics in shaping the healthcare experience. The framework informed our interpretation of the identified factors influencing satisfaction with care. When examining each factor, we considered the role of gender and its intersections on women’s healthcare expectations and perceptions.
- It would be helpful to know whether the synthesis of the literature review were done with the assistance of the 2 RAs who were community partners. If not, that may be a limitation where the RAs were involved later on.
Response: Thank you for this comment. We have now acknowledged in our limitations that the study PRAs did not participate in the literature review synthesis.
Our literature review synthesis and construction of the literature-review map did not include PRA perspectives. Through mapping interviews, we aimed to integrate the perspectives of women living with HIV in existing literature.
Reviewer 2 Report
First of all, the authors are to be congratulated on their work. I believe that the subject they have chosen is of crucial importance today for diversity and wellbeing.
Methods section needs to be added some words to be easy to understand.
Firstly, how is "2.1. Theoretical framework, add the figure if necessary, what is
2.2. Participatory research approach" and what is 2.3. Fuzzy Cognitive Mapping in this work,
Author Response
First of all, the authors are to be congratulated on their work. I believe that the subject they have chosen is of crucial importance today for diversity and wellbeing.
Response: Thank you for your comments acknowledging the relevance of this study for patient care and wellbeing.
Methods section needs to be added some words to be easy to understand. Firstly, how is
2.1. Theoretical framework, add the figure if necessary, what is
Response: We have revised the theoretical framework section to clarify the role of the theoretical framework in informing our study aims, design, and interpretations.
2.2. Participatory research approach"
Response: We have revised the Participatory research approach section to include a definition of participatory research. The following text has been added
We applied a participatory research approach, which aims to meaningfully involve community members in research impacting their community by responding to their priorities and drawing on their strengths and knowledge [18].
and what is 2.3. Fuzzy Cognitive Mapping in this work,
Response: Thank you for your comment. We have revised the section Fuzzy Cognitive Mapping to highlight the application of this method in our study. The following details were added:
FCM is a systematic process of knowledge creation that generates a concept map representing stakeholder understanding of causal relationships …
In this study, we used FCM to collate different sources of causal understanding, namely literature evidence and stakeholder knowledge.